# Out-of-pocket payments and associated factors among hypertensive patients insured under the National Health Insurance Scheme in a referral hospital, Ghana

**Angela Nana Esi Ackon**[1]*, **Hubert Amu**[2], **Martin Amogre Ayanore**[1]

**1** Department of Health Policy Planning and Management, Fred N. Binka School of Public Health, University of Health and Allied Sciences, Hohoe, Ghana, **2** Department of Population and Behavioural Sciences, Fred N. Binka School of Public Health, University of Health and Allied Sciences, Hohoe, Ghana

* angela.aan2009@gmail.com

## Abstract

### Introduction

Out-of-pocket (OOP) payments are a common component of healthcare financing in many low-and middle-income countries including Ghana. Despite implementation of the National Health Insurance Scheme in Ghana which is expected to protect against catastrophic OOP payments, patients still make some form of payments with its attendant negative implications for improved health status. We examined the prevalence and self-reported effects of OOP payments among hypertensive patients insured under the National Health Insurance Scheme attending a referral hospital in Hohoe, Ghana.

### Methods

This was a cross-sectional study conducted among 389 hypertensive patients between February and May 2022 using a structured questionnaire. Data collected were analysed using Stata software version 15. We employed descriptive statistics and logistic regression in analysing the data. Statistical significance was considered at $p < 0.05$.

### Results

A total of 97.2% respondents made OOP payments in receiving healthcare with 97.1% of those who made such payments having made them at the point of receiving drugs. Difficulty in making OOP payments was reported by 29.1% of the respondents and this was associated with ever borrowing money to cover health expense (aOR=2.52; 95% CI: 1.31–4.88; $p = 0.006$), OOP payments affecting hospital

**Data availability statement:** All relevant data are within the paper and its Supporting Information files.

**Funding:** The author(s) received no specific funding for this work.

**Competing interests:** The authors have declared that no competing interests exist.

**Abbreviations:** CNCD, Chronic non-communicable disease; CVD, Cardiovascular disease; LMIC, Low-and middle-income country; NCD, Non-communicable disease; NHIA, National Health Insurance Authority; NHIS, National Health Insurance Scheme; O&G, Obstetrics and Gynaecology; OOP, Out-of-pocket; OPD, Out-patient-department; PI, Principal Investigator; REC, Research Ethics Committee; SDG, Sustainable Development Goal; SSA, Sub-Saharan Africa; UHAS, University of Health and Allied Sciences; UHC, Universal Health Coverage; WHO, World Health Organization.

attendance (aOR=5.52; 95% CI: 2.74–11.11; $p < 0.001$) and having an alternative form of treating hypertension (aOR=2.93; 95% CI: 1.60–5.38; $p = 0.001$).

## Conclusion

OOP payments are widespread in the utilisation of healthcare for hypertensive patients. This raises concerns about progress towards the attainment of Sustainable Development Goals (SDGs) 3.4 which seeks to control non-communicable diseases and SDG 3.8 which seeks to ensure universal health coverage. Medications or drugs are the major healthcare services for which individual hypertensive patients made OOP payments thus highlighting the need ensure availability and accessibility of drugs to meet the global targets of SDG 3 and due to the long-term treatment requirements of hypertension.

## Introduction

Out-of-pocket (OOP) payments lead to financial and non-financial barriers in healthcare access [1] and such payments are a common component of healthcare financing in low- and middle-income countries (LMICs). OOP expenditure by households is reported as a regressive pathway to financing healthcare since it leads to the risk of catastrophic payments by households who do not have adequate financial protection [2]. OOP payments refer to payments, both formal and informal, that are made while accessing healthcare and these include payment for medical supplies, consultation fees and payment for laboratory tests [1]. It also includes any direct remittance by households to healthcare providers which is primarily intended to restore or enhance health of an individual or group [3].

The establishment of the National Health Insurance Scheme (NHIS), a public health financing scheme which delivers accessible, affordable, and good quality health care to all Ghanaians especially the poor and most vulnerable in society [4], is an important pathway towards Ghana's ambition to achieve Universal Health Coverage (UHC) [5,6].

Nevertheless, patients seeking health services are confronted with payments such as user fees, payment for medicines, consultation fees, and informal fees, which create a huge gap in achieving UHC in Ghana and other LMICs [7]. OOP payments are common at all levels of health facilities in the country with about 40–53% of patients making payments and approximately 47% of Ghana's NHIS clients with valid cards reportedly making OOP payments for out-patient-department (OPD) services [8]. Although subscription to the NHIS has reduced household OOP payments by 86%, some insured households still make OOP payments [2,7].

People with chronic conditions need follow-up visits and long-term use of medications making them more prone to OOP payments and invariably catastrophic health expenditure [9] with chronic non-communicable diseases (CNCDs) becoming increasingly burdensome to health care systems worldwide and about 71% of all

global deaths attributable to CNCDs [10]. Hypertension remains one of the major CNCDs which also serves as a risk factor for the development of cardiovascular disease (CVD) [11].

The burden of hypertension among Ghanaians is high, with one out of every four persons having the condition [12]. Among older adults, as many as 53.7% were hypertensive [13].

CNCDs including hypertension require considerable financial resources to manage. The costs involved include medications and transportation costs for hospital reviews [14]. Ghana's NHIS is expected to cover over 95% of disease conditions that afflict Ghanaians as stated in the policy framework that established the scheme [15] and this includes the management of hypertension. The categories of medications for treatment of hypertension are quite common for persons with the condition in Ghana.

A challenge, however, is the unavailability of these essential drugs at lower-level service delivery points [16]. Also, not all the drugs are listed on the NHIS medicines list and the ones listed for hypertension are usually not accessible to patients due to frequent drug shortages in many medical stores in the country [17]. This will invariably compel the hypertensive patient to make OOP payments at the point of healthcare utilisation. OOP payments have brought poverty and financial disaster upon many individuals and households in many countries with about 100 million persons being pushed below the poverty line globally due to healthcare utilisation and its associated OOP payments which leads to persons not seeking required health services due to financial circumstances and this has a ripple effect where they suffer due to ill health, lose their jobs and sink further into poverty [18,19].

The Sustainable Development Goals (SDGs) represent a clarion call for every member of the United Nations to commit to peace and prosperity of all their citizens [20]. The third goal of the SDGs primarily seeks to ensure and promote health and wellbeing for everyone irrespective of age with achieving UHC and the control of non-communicable diseases (NCDs) as some of the key targets under SDG 3 [21]. OOP payments however threaten the attainment of UHC under SDG 3.8 and OOP payments particularly among hypertensive patients subscribed to health insurance dim the prospects of attaining SDG 3.4.

Earlier studies on OOP payments in Ghana have mainly sought to examine the impact of the NHIS on OOP payments and catastrophic health expenditure [2,7,8,22,23]. However, it is also imperative to determine the difficulties that insured individuals undergo when they are faced with OOP payments. Given the increasing reports of OOP payments at various health facilities in Ghana [2,8,24] and considering the long-term healthcare requirement of hypertension, a study on OOP payments among hypertensive patients who are subscribed to the NHIS was needed. In this paper, our contribution to the literature on OOP payments determines the prevalence and self-reported effects of making OOP payments by hypertensive patients who are subscribed to the NHIS.

## Methods

Our study followed the 'Strengthening the Reporting of Observational studies in Epidemiology' (STROBE) directive guidelines for observational studies [25].

### Study design

This is a health facility-based cross-sectional study. This study design was employed due to its ability to examine multiple exposure variables of an outcome among a study population [26].

### Study settings

The study was conducted between February 2022 and May 2022 at the Volta Regional Hospital, located in the Hohoe Municipality of the Volta Region, Ghana. Hohoe is one of the 18 districts/municipalities in the Volta Region and is in the northern part of the region. The municipality shares boundaries on the north with Jasikan District, Biakoye District on the

north-west, Kpando Municipality on both west and south-west, Afadjato South District on the south and on the east with the Republic of Togo. The municipality consists of one hundred and two communities with an estimated 2021 population of 216,038 based on the 2010 national population census at an annual growth rate of 2.5%. Major economic activities include farming (about 55%), trading (about 25%), livestock rearing (about 15%) and others (about 5%). The municipality has been divided into 7 sub-municipalities namely Akpafu/Santrokofi, Alavanyo, Agumatsa, Lolobi Gbi-South, Hohoe-Sub and Likpe. It has a total of twenty-six (26) health institutions including the Volta regional hospital, a polyclinic, a research centre, 19 health centres, a private hospital and 33 CHPS zones. There is also a midwifery training school which is a diploma awarding institution in midwifery [27].

The Volta Regional Hospital, Hohoe where the study was conducted, was established in 1940 and has been upgraded to the status of a regional hospital upon the conversion of the former Volta regional hospital, Ho into a teaching hospital [28]. The facility is led by a medical director with support from the various heads of units. Essential units within the hospital include the Obstetrics and Gynaecology unit (O&G), the OPD, the administration, the physiotherapy unit among others. CNCD cases are typically managed at the OPD and the hypertension clinic. The hospital is an NHIS accredited facility and provides general services.

### Inclusion and exclusion criteria

All hypertensive patients accessing care at the Volta regional hospital, Hohoe and insured under the NHIS were included in the study. The exclusion of potential respondents in this study was based on all those who met the inclusion criteria but were seriously ill at the time of the data collection, not of sound mind, or did not attend the hospital during the period of data collection. For the purposes of this study, a hypertensive patient is a person aged 18 years and above with either systolic blood pressure >140 mmHg or diastolic blood pressure >90 mmHg or those taking anti-hypertensive treatment regardless of their blood pressure on measurement [12].

Hypertensive patients who subscribed to the NHIS were selected using a systematic random sampling technique. Approximately 80 patients visited the hypertensive clinic at each clinic session at the Volta regional hospital and 60 hypertensive patients were systematically selected each clinic day to participate in the study for a period of two months until the required sample size of hypertensive patients was obtained. A balloting without replacement method was used for randomisation. "Yes" and "No" were written on pieces of paper and then folded. The folded pieces were then placed in a basket and tumbled to ensure a good mix. In the situation where patients visiting the hypertensive clinic were less than 80, all eligible patients present were allowed to participate in the study after consenting voluntarily. To ensure that a respondent did not participate more than once in this study, prospective respondents were asked after data collection had taken place for four weeks if they had met the data collection team. This was because the patients had a usual review schedule of monthly and once every two months. In addition, phone numbers of the respondents were collected and checked for duplicates as a double measure.

### Study variables

**Outcome variables.** In our study, the primary outcome variable is making OOP payments which indicates the payment of money for accessing healthcare at the hospital. This was determined by posing the question "Were you made to pay for any of the services?" to the respondents with responses as either a "yes" or "no". The preceding question sought to determine which health services the respondent assessed and followed up with whether payments had been made for any of those payments, providing a detailed breakdown of possible health services that may have resulted in payments. Informal payments and co-payments were included. Our study, however, excluded payments made outside of the health facility since we sought to examine the OOP payments made by NHIS subscribers. It was coded "1" if a respondent reported having been asked to make any payment and "0" if otherwise.

The secondary outcome variable is a binary variable which measures the self-reported difficulty that arises due to having made OOP payments for healthcare utilisation. This was a follow-up question for respondents who made OOP payments. It was determined by the question "Did you have difficulty making these payments?" with responses as either a "yes" or "no". It was coded "1" if a respondent reported having difficulties in making OOP payments and "0" if otherwise.

## Explanatory variables

The explanatory variables fell under two categories: namely socio-demographic variables and the contextual factors of making OOP payments. The socio-demographic variables consisted of age, sex, religion, marital status, educational level, occupation, form of income and level of income earned. On the other hand, service for which OOP payments were made, amount of money paid as OOP payments, source of money used to pay for OOP payments, reasons for making OOP payments, ever borrowing money to attend the hospital, self-reported effects of OOP payments on managing hypertension and alternative means of managing hypertension were considered the contextual variables.

## Sampling

**Sample size determination.** Using the cross-sectional study formula [29] with an expected OOP payment prevalence of 47% [8] and assuming a z-statistic for a 95% level of confidence and a 5% margin of error, the appropriate minimum sample size was estimated for the study as:

$$n = \frac{Z^2(1-p)p}{d^2}$$

Where:

n = sample size

Z = Confidence interval of 95% (standard value of 1.96)

P = expected prevalence or proportion; 0.47 [8]

d = precision: 5% margin of error; 0.05

Considering these assumptions, the actual sample size for the study was calculated using the formula:

$n = \frac{(1.92^2) \times 0.47 \times (1-0.47)}{0.05^2}$

n = 382.77

Adjusting for non-response rate of 10% of 383 = 422

This study however recruited 393 respondents due to practical recruitment constraints

## Data collection

We used a structured questionnaire to collect data for this study. The questionnaires were in English but in instances where a respondent did not understand English, it was translated into either Ewe or Twi by a local translator. The administration of each questionnaire took at least 25 minutes.

There were four sections on the questionnaire namely; A, B, C, and D. Sociodemographic information on the respondents was recorded in Section A. Data on the proportion of hypertensive patients subscribed to the NHIS who had made OOP payments was captured in Section B. Section C captured the factors influencing OOP payments and Section D contained questions on the perceived influence of OOP payments on respondents' accessing healthcare.

Each questionnaire was assigned a unique code before data collection. At the end of the field work, a secure database was created to contain all the information about the study respondents. Pre-testing was carried out in the Ho municipality of the Volta Region to fine-tune the questions. Data were collected using KoBo Toolbox v2022.1.2 software on a

face-to-face basis with strict adherence to COVID-19 protocols. The data collection for this study started on the 8th of February 2022 and ended on the 7th of May 2022. All OOP payments and income in the study were collected in GHC, and the results are presented in GHC with the USD equivalent in brackets, using the average exchange rate of May 2022 when data collection ended (USD 1 = GHC 7.13) [30].

## Data analysis

Data was exported from Kobo Toolbox into Microsoft Excel 2016 for cleaning and validation to ensure quality before analysis began. Cleaning of the data was done by running frequencies of the variables to check for inconsistencies in data coding. For analysis, the cleaned data were exported to STATA Windows version 15.0. Descriptive statistics were used to describe respondents' socio-demographic characteristics, the proportion of respondents who made OOP payments, reasons for making OOP payments and the self-reported effects of OOP payments on their health status and demand for healthcare. The results are presented as proportions and means and are displayed in tables, pie chart and graphs. Pearson Chi-square test or Fisher's exact test (where expected cell count is < 5) was used to determine the association between making OOP payments and socio-demographic variables. Multiple logistic regression was used to determine factors that influenced difficulty in making OOP payments with statistical significance considered based on a p-value of <0.05 at a confidence interval of 95%. The results are presented as odds ratios.

## Ethical considerations

This study received ethical approval from the University of Health and Allied Sciences' (UHAS) Research Ethics Committee (REC); UHAS REC Protocol identification number UHAS-REC A.3 [3] 21–22. Permission and approval were obtained from the medical director of the Volta regional hospital. Consent to participate in the study was also obtained through written informed consent forms from the participants. The consent form contained information on the objectives of the study, risks, benefits and freedom of participation, and confidentiality.

## Results

### Socio-demographic characteristics of respondents

Table 1 presents socio-demographic characteristics of the respondents. The number of questionnaires administered was 393 but 4 were incomplete and were therefore not included in the analysis. The response rate was thus 99%. Females represented 82.3% of the respondents. The mean age of the respondents was 66 ± 11.87 years. Only 5.9% of the respondents were below 50 years old and the rest were above 50 years. Based on religious affiliation, a greater proportion of the participants (93.3%) were Christians. With respect to marital status, 65.3% of the respondents were not married. In addition, less than half of the respondents (20.6%) had no form of education and the rest had some form of education. With regards to occupation, 26.7% of the respondents were unemployed, 16.4% were retired, 27.8% were into sales and services, 26.0% were into agriculture or manual work and 3.1% were clerical workers. In addition, more than half of the respondents (58.2%) earned less than GH¢500.00 (equivalent of USD70.13) as monthly income and the rest earned GH¢500.00 (USD70.13) and above.

### Prevalence of OOP payments

The number of respondents who indicated having been asked to make OOP payments at the hospital was 97.2% and the remaining 2.8% indicated that they did not make any OOP payments. This is illustrated in Fig 1 below.

### Socio-demographic factors associated with making OOP payments

More females (82.8%) made OOP payments than males (17.2%). Additionally, only 6.1% of respondents who made OOP payments were less than 50 years old, with the rest aged 50 years and above. Similarly, a greater proportion (93.9%) of

**Table 1. Socio-demographic characteristics of respondents.**

| Variable | Frequency (n = 389) | Percent (%) |
|---|---|---|
| Mean age (SD) | 66 (11.87) | |
| **Age (in years)** | | |
| <50 | 23 | 5.9 |
| 50-59 | 100 | 25.7 |
| 60-69 | 109 | 28.0 |
| 70-79 | 89 | 22.9 |
| 80+ | 68 | 17.5 |
| **Sex** | | |
| Male | 69 | 17.7 |
| Female | 320 | 82.3 |
| **Religion** | | |
| Christianity | 363 | 93.3 |
| Islam | 26 | 6.7 |
| **Marital status** | | |
| Separated/Never married | 29 | 7.5 |
| Married | 174 | 44.7 |
| Widowed | 153 | 39.3 |
| Divorced | 33 | 8.5 |
| **Highest level of education** | | |
| None | 80 | 20.6 |
| Primary | 85 | 21.9 |
| Junior High | 159 | 40.9 |
| Tertiary/Senior High | 65 | 16.7 |
| **Occupation** | | |
| Unemployed | 104 | 26.7 |
| Manual | 101 | 26.0 |
| Clerical/Managerial | 12 | 3.1 |
| Retired | 64 | 16.4 |
| Sales & services | 108 | 27.8 |
| **Income level** | | |
| Less than GH¢500 | 226 | 58.1 |
| GHC500–999 | 99 | 25.5 |
| GHC1000 and above | 64 | 16.4 |

those who made OOP payments were Christians. With respect to marital status, almost half (45%) of the respondents who made OOP payments were married. Further, only 20.4% of those who made OOP payments had no form of education, with the rest having some education. With regards to occupation, 26.7% of those who made OOP payments were unemployed and the rest were in some form of employment. Based on the amount of income earned, more than half (58.2%) of the respondents who made OOP payments earned less than GHC500.00 (equivalent of USD70.13) as monthly income and the rest earned GHC500.00 (USD70.13) and above. In addition, 61.1% of the respondents who made OOP payments visit the hospital every two months. A significant association was found between religion of respondents and making OOP payments ($\chi^2$=7.69, p = 0.006), (see Table 2).

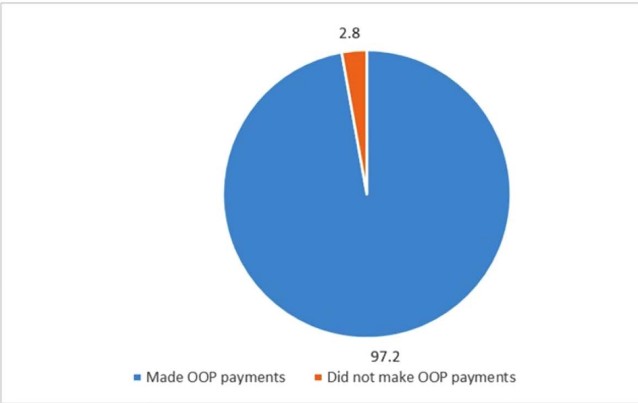

**Fig 1. Proportion of respondents who made OOP payments.**

### OOP payments among respondents

Table 3 presents details on the OOP payments made by respondents in accessing healthcare. A greater proportion of the respondents (97.1%) who had made OOP payments indicated that they made OOP payments in relation to drugs that they were given, 7.4% reportedly made OOP payments in accessing laboratory services, 2.4% each with regards to X-ray and OPD services and 3.7% made these payments in accessing other services. A greater proportion (93.4%) of respondents who made OOP payments paid less than GHC50 (USD7.01), 3.4% and 3.2% made payments between GHC50 (USD7.01) to GHC99 (USD13.88) and GHC100 (USD14.03) to GHC199 (USD27.91) respectively. More than half of the study participants (57.7%) indicated that the monies they used to cover OOP costs were from personal savings, 38.6% reported that their children provided them with the money, 6.6% indicated that it was their partners and 2.9% indicated that their family members provided them with the money that they used to make OOP payments. In addition, 29.1% of those who made OOP payments indicated that they had difficulties in their willingness and ability to make the payments and 70.9% of those who made OOP payments indicated that they had no difficulty making the payments.

### Reasons for making OOP payments

Fig 2 illustrates the views of the respondents on the reasons why they made OOP payments despite being subscribers to NHIS. More than half (74.6%) of respondents expressed ignorance as to the reasons why they had to make extra payments, 6.4% of respondents reported that it was due to low insurance coverage and 19.1% indicated that it was due to shortage of drugs.

### Self-reported effects of OOP payments on respondents

Table 4 presents the self-reported effects of OOP payments on respondents. A greater proportion of the survey respondents (84.1%) reported that they had never had to borrow money to pay hospital bills and 15.9% of them indicated that they had borrowed money. More than half (79.0%) of the respondents who had ever borrowed money to be able to pay hospital bills indicated that they borrowed less than GHC50 (USD7.01), 11.3% of them borrowed amounts between GHC50 (USD7.01) to GHC99 (13.88) and 9.7% borrowed amounts between GHC100 (USD14.03) to GHC199 (USD27.91). With respect to OOP payments limiting financial abilities concerning other expenses, 71.5% of the survey respondents noted that OOP payments did not limit their financial ability concerning other expenses and 28.5% reported that OOP payments indeed limited their financial abilities concerning other expenses. The respondents who affirmed that

**Table 2. Socio-demographic characteristics associated with OOP payments.**

| Variables | No OOP payments(%) | Made OOP payments(%) | Chi² (p-value) |
|---|---|---|---|
| **Age (in years)** | | | |
| <50 | 0(0.0) | 23(6.1) | 6.52(0.141)** |
| 50-59 | 0(0.0) | 100(26.5) | |
| 60-69 | 6(54.5) | 103(27.2) | |
| 70-79 | 3(27.3) | 86(22.7) | |
| 80+ | 2(18.2) | 66(17.5) | |
| **Sex** | | | |
| Male | 4(36.4) | 65(17.2) | 2.69(0.112)** |
| Female | 7(63.6) | 313(82.8) | |
| **Religion** | | | |
| Christianity | 8(72.7) | 355(93.9) | **7.69(0.006)*** |
| Islam | 3(27.3) | 23(6.1) | |
| **Marital status** | | | |
| Separated/Never married | 0(0.0) | 29(7.7) | 3.68(0.495)** |
| Married | 4(36.4) | 170(45.0) | |
| Widowed | 7(63.6) | 146(38.6) | |
| Divorced | 0(0.0) | 33(8.7) | |
| **Highest level of education** | | | |
| None | 3(27.3) | 77(20.4) | 4.36(0.184)** |
| Primary | 2(18.2) | 83(22.0) | |
| Junior High | 2(18.2) | 157(41.5) | |
| Tertiary/Senior High | 4(36.4) | 61(16.1) | |
| **Occupation** | | | |
| Unemployed | 3(27.3) | 101(26.7) | 9.02(0.061)** |
| Agriculture/Manual | 2(18.2) | 99(26.2) | |
| Clerical/Managerial | 2(18.2) | 10(2.7) | |
| Retired | 2(18.2) | 62(16.4) | |
| Sales & services | 2(18.2) | 106(28.0) | |
| **Income level** | | | |
| Less than GHC500 | 6(54.5) | 220(58.2) | 0.06(0.970)** |
| GHC500–999 | 3(27.3) | 96(25.4) | |
| GHC1000 and above | 2(18.2) | 62(16.4) | |
| **Frequency of hospital attendance** | | | |
| Weekly/fortnight | 0(0.0) | 10(2.7) | 0.33(1.000)** |
| Monthly | 2(18.2) | 74(19.6) | |
| Every two months | 7(63.6) | 231(61.1) | |
| Three months and above | 2(18.2) | 63(16.7) | |

\* Statistically significant at a p-value< 0.05.

\*\* Fisher's exact.

OOP payments limited their financial abilities indicated food, clothing, rent, business capital and utility bills as some of the items that were negatively affected by their making OOP payments at the point of accessing health care.

More than half of the study respondents (78.6%) indicated that their hospital attendance was not affected by the OOP payments that they were made to pay thus reported no change in hospital attendance and 21.4% of the participants

**Table 3. OOP payments among respondents.**

| Variable | Frequency (n = 378) | Percent (%) |
|---|---|---|
| **Service for which patient made OOP payments** | | |
| OPD services | 9 | 2.4 |
| Drugs | 368 | 97.1 |
| Laboratory service | 28 | 7.4 |
| X-ray | 9 | 2.4 |
| Others | 14 | 3.7 |
| **OOP amount** | | |
| Less than GHC50 | 353 | 93.4 |
| GHC50- GHC99 | 13 | 3.4 |
| GHC100- GHC199 | 12 | 3.2 |
| **Source of funding for OOP payments** | | |
| Self | 218 | 57.7 |
| Partner | 25 | 6.6 |
| Children | 146 | 38.6 |
| Family members | 11 | 2.9 |
| **Difficulty making OOP payments** | | |
| Yes | 110 | 29.1 |
| No | 268 | 70.9 |

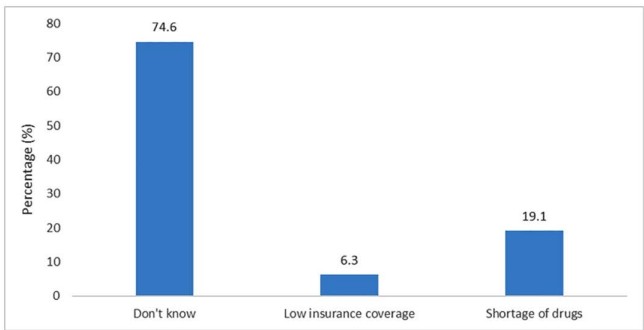

**Fig 2. Reasons for making OOP payments.**

reported that the OOP payments made them either anxious or reluctant about their next visit to the hospital with some reporting that the OOP payments made them not to attend the hospital again as presented in Table 3.

With regards to whether the thought of making extra payments influenced their hospital attendance, a greater proportion of respondents (79.4%) indicated that OOP payments did not influence their hospital attendance and 20.6% indicated that hospital attendance was influenced by OOP payments. In addition, a greater proportion of study participants (90.0%) indicated that OOP payments had no effect on the management of their conditions based on their perspective and 9.0% reported that OOP payments worsened their condition.

With regards to having alternative treatment plans for managing hypertension, more than half of the study participants (81.2%) indicated the scheduled hospital visits as their and 18.8% indicated that they had alternative means of managing hypertension. The common alternative means the respondents mentioned were herbs (see Table 4).

**Table 4. Self-reported effects of OOP payments by study participants.**

| Variable | Frequency (n=389) | Percent (%) |
|---|---|---|
| **Ever borrowed money to attend hospital** | | |
| Yes | 62 | 15.9 |
| No | 327 | 84.1 |
| **Amount of money borrowed to attend hospital** | | |
| Less than GHC50 | 49 | 79.0 |
| GHC50- GHC99 | 7 | 11.3 |
| GHC100- GHC199 | 6 | 9.7 |
| **OOP payments limiting financial ability** | | |
| Yes | 111 | 28.5 |
| No | 278 | 71.5 |
| **Does OOP payments influence hospital attendance** | | |
| Yes | 80 | 20.6 |
| No | 209 | 79.4 |
| **Effect of OOP payments on subsequent hospital visit** | | |
| No change in hospital attendance | 306 | 78.6 |
| Anxious about next hospital visit | 10 | 2.6 |
| Reluctant to visit hospital | 68 | 17.5 |
| Not come to the hospital again | 5 | 1.3 |
| **Self-reported effect of OOP payments on hypertension** | | |
| Worsened condition | 35 | 9.0 |
| No effect on condition | 354 | 91.0 |
| **Alternative means of managing hypertension** | | |
| Yes | 73 | 18.8 |
| No | 316 | 81.2 |

### Multivariable analysis of factors influencing difficulty in making OOP payments

Table 5 presents factors influencing difficulty in making OOP payments. Marital status was the only socio-demographic factor that was significantly associated with having difficulty in making OOP payments before adjustment. Married hypertensive patients subscribed to the NHIS were 61% less likely to have difficulty in making OOP payments compared to non-married hypertensive patients (cOR= 0.39; 95% CI: 0.17–0.88; p=0.023). Respondents for whom OOP payments limited their financial ability in terms of other expenditure were three times more likely to have difficulty in making OOP payments than those that OOP payments did not limit their financial ability before adjustment (cOR=3.13; 95% CI: 1.95–5.03; p<0.001).

After adjustments, ever borrowing money to cover health expenses, OOP payments affecting hospital attendance and having an alternative form of treating hypertension were significantly associated with having difficulty making OOP payments. The respondents who had ever borrowed money to cover health expenses were two times more likely to have difficulty in making OOP payments compared to participants who had never borrowed money to cover their health expenses (aOR=2.52; 95% CI: 1.31–4.88; p=0.006).

Additionally, respondents who indicated that OOP payments influenced their hospital attendance were five times more likely to have difficulty in making OOP payments compared to those for whom OOP payments had no influence on their hospital attendance (aOR=5.52; 95% CI: 2.74–11.11; p<0.001) and study participants who had an alternative method of treating hypertension such as herbs were twice more likely to have a difficulty in making OOP payments than those who

**Table 5. Factors associated with difficulty in making OOP payments.**

| Variable | Difficulty in making OOP payments | | cOR(95% CI) p-value | aOR(95% CI) p-value |
|---|---|---|---|---|
| | No (%) | Yes (%) | | |
| **Background characteristics** | | | | |
| **Age (in years)** | | | | |
| <50 | 16(6.0) | 7(6.4) | Ref | |
| 50-59 | 74(27.6) | 26(23.6) | 0.80(0.30,2.17)0.666 | |
| 60-69 | 71(26.5) | 32(29.1) | 1.03(0.39,2.75)0.953 | |
| 70-79 | 52(19.4) | 34(30.9) | 1.49(0.56,4.01)0.425 | |
| 80+ | 55(20.5) | 11(10.0) | 0.46(0.15,1.37)0.163 | |
| **Sex** | | | | |
| Female | 218(81.3) | 95(86.4) | Ref | |
| Male | 50(18.7) | 15(13.6) | 0.69(0.37,1.29)0.242 | |
| **Religion** | | | | |
| Christianity | 245(91.4) | 110(100.0) | | |
| Islam | 23(8.6) | 0(0.0) | Omitted | |
| **Marital status** | | | | |
| Separated/Never married | 16(6.0) | 13(11.8) | Ref | Ref |
| Married | 129(48.1) | 41(37.3) | 0.39(0.17,0.88)0.023* | 0.63(0.25,1.58)0.321 |
| Widowed | 105(39.2) | 41(37.3) | 0.48(0.21,1.09)0.078 | 0.67(0.27,1.71)0.405 |
| Divorced | 18(6.7) | 15(13.6) | 1.03(0.38,2.80)0.961 | 0.85(0.27,2.65)0.778 |
| **Highest level of education** | | | | |
| None | 56(20.9) | 21(19.1) | Ref | |
| Primary | 64(23.9) | 19(17.3) | 0.79(0.39,1.62)0.523 | |
| Junior High | 98(36.6) | 59(53.6) | 1.61(0.88,2.92)0.120 | |
| Tertiary/Senior High | 50(18.7) | 11(10.0) | 0.59(0.26,1.34)0.204 | |
| **Occupation** | | | | |
| Unemployed | 75(28.0) | 26(23.6) | Ref | |
| Agriculture/Manual/ | 64(23.9) | 35(31.8) | 1.58(0.86,2.90)0.141 | |
| Clerical/Managerial | 10(3.7) | 0(0.0) | 1 | |
| Retired | 40(14.9) | 22(20.0) | 1.59(0.80,3.15)0.187 | |
| Sales & services | 79(29.5) | 27(24.6) | 0.99(0.53,1.84)0.964 | |
| **Form of income earned** | | | | |
| Cash & kind | 92(34.3) | 41(37.3) | Ref | |
| Cash only | 66(24.6) | 31(28.2) | 1.05(0.60,1.85)0.855 | |
| In kind only | 12(4.5) | 8(7.3) | 1.50(0.57,3.94)0.414 | |
| Not paid | 98(36.6) | 30(27.3) | 0.69(0.40,1.19)0.181 | |
| **Income level** | | | | |
| Less than GHC500 | 158(59.0) | 62(56.4) | Ref | |
| GHC500–999 | 66(24.6) | 30(27.3) | 1.16(0.69,1.95)0.581 | |
| GHC1000 and above | 44(16.4) | 18(16.3) | 1.04(0.56,1.94)0.896 | |
| **Contextual characteristics** | | | | |
| **Amount of OOP cost paid** | | | | |
| Less than GHC50 | 255(95.2) | 98(89.1) | Ref | |
| GHC50–99 | 7(2.6) | 6(5.5) | 2.23(0.73,6.80)0.159 | |
| GHC100–199 | 6(2.2) | 6(5.5) | 2.60(0.82,8.26)0.105 | |

*(Continued)*

**Table 5.** (Continued)

| Variable | Difficulty in making OOP payments | | cOR(95% CI) p-value | aOR(95% CI) p-value |
|---|---|---|---|---|
| **Ever borrowing money to cover health expense** | | | | |
| No | 234(87.3) | 82(74.6) | Ref | Ref |
| Yes | 34(12.7) | 28(25.5) | 2.35(1.34,4.11)0.003* | 2.52(1.31,4.88)0.006* |
| **OOP payments limiting financial ability** | | | | |
| No | 210(78.4) | 59(53.6) | Ref | Ref |
| Yes | 58(21.6) | 51(46.4) | 3.13(1.95,5.03)<0.001* | 0.90(0.46,1.76)0.761 |
| **OOP payments influencing hospital attendance** | | | | |
| No | 237(88.4) | 61(55.5) | Ref | Ref |
| Yes | 31(11.6) | 49(44.6) | 6.14(3.61,10.44)<0.001* | 5.52(2.74,11.11)<0.001* |
| **Alternative form of treating hypertension** | | | | |
| No | 231(86.2) | 76(69.1) | Ref | Ref |
| Yes | 37(13.8) | 34(30.9) | 2.79(1.64,4.76)<0.001* | 2.93(1.60,5.38)0.001* |

* Statistically significant at $p < 0.05$.

did not have any other method of treating hypertension apart from scheduled hospital visits (aOR=2.93; 95% CI: 1.60–5.38; p=0.001), (see Table 5).

## Discussion

This study sought to measure OOP payments among hypertensive patients insured under the NHIS. Our results observed a significantly higher prevalence of OOP payments in Ghana as compared to a previous study in the country [8]. This may be explained by the homogenous nature of our respondents which is further confirmed by another homogenous study on maternal health which reported similar prevalence of OOP payments as our findings [31]. Based on the results of this study, 97.2% of study respondents made OOP payments to access healthcare services and 2.8% did not make any OOP payments. This is corroborated by a previous study which reported a 93.6% OOP expense incurred as a result of seeking healthcare for hypertension in a low-income urban part of Columbia [32]. These results imply that managing hypertension is costly. Although 93.4% of the respondents paid amounts less than GHC50 (USD7.01), it is important that this is not discounted. This is because based on the socio-demographic characteristics of the respondents in this study, the average age was 66 years, almost half of them (43.1%) were either retired or unemployed, 58.1% earned income below GHC500.00 (USD70.13) monthly and were required to attend scheduled periodic hospital reviews thereby incurring further costs. Our findings are reiterated by a study in Ghana which emphasized that having health insurance albeit important, may not provide adequate financial protection for the poor [22].

A similar study also revealed that no single individual had zero OOP expense although the authors suggest that OOP expenditure is reduced considerably by having insurance [33]. Results from a similar study also confirm that OOP payments are fast becoming a reality on all kinds of health services including those covered by insurance for both in-patient and out-patient services [2]. In Taiwan, research has identified that the country's National Health Insurance system successfully reduced household OOP payments by about 23% though it could not fully ease the financial distress caused by sickness with households in the lower- and middle-income quintiles experiencing only minimal reductions in OOP payments [34] and in Nigeria, activism for OOP health expenditure has relented as proponents agree that it negatively affects the poor [35].

Based on the findings of this study, drugs alone accounted for 97.1% of health services on which patients made OOP payments. This finding is corroborated by a previous study which indicated that pharmaceuticals make up a significant

proportion of healthcare expenditure in many countries [34]. Another study in Peru revealed that a greater proportion of incurred OOP expenditure was attributable to medications [36]. This is confirmed by research in Chile which indicated that drugs are the major components of OOP payments [37]. This is consistent with another research on maternal health in northern Ghana which established that patients made OOP payments as a result of drug stock-outs [38]. Another study indicated that OOP payments were attributable to shortage of prescribed drugs or prescription of drugs not on the NHIS drug list and this was as a result of constant delays in reimbursement by the NHIS to health service providers invariably leading to either periodic shortages or refusal of service providers to provide NHIS subscribers with the needed drugs [8].

Data gathered in this study revealed that laboratory services accounted for 7.4% of OOP payments with X-ray and OPD services accounting for 2.4% each. In essence, OOP payments and the resultant financial burden differ with respect to the type of health service utilized with more than 92% of those who access out-patient services at private health facilities spending money on OOP payments and about 70% of attendants of public health facilities spending on OOP costs related to in-patient care services and a considerably larger percentage in cases of surgery [39]. In Bangladesh, research has identified that the average cost of disease-specific OOP payments was considerably higher in cases of chronic illnesses [40].

Based on the results of the study, 29.1% of the survey respondents who had made OOP payments expressed difficulty in their willingness and ability to make OOP payments. Overall, this study did not find significant association between difficulty in making OOP payments and socio-demographic factors except marital status before adjustment. A similar study on maternal health in Ghana however established that women from low neighbourhood disadvantage levels had a higher prevalence of OOP payment [18].

A factor that influenced difficulty in making OOP payments was ever borrowing money to cover health expenses as results from the survey respondents indicated. Respondents who had ever borrowed money to cover health expenses were two times more likely to have difficulty in making OOP payments compared to participants who had never borrowed money to cover their health expense. This is consistent with findings from a study in Zambia which established that persons who were faced with unaffordable OOP payments experienced hardship financing which was characterized by borrowing money for healthcare [41]. A similar study in Cambodia revealed that the households incurred an average of US$125 due to borrowing in order to access healthcare and experienced hardship financing [42].

In this study, respondents for whom OOP payments limited their financial ability in terms of other expenditure were three times more likely to have difficulty in making OOP payments than those that OOP payments did not limit their financial ability before adjustment. This is consistent with findings on SSA which reported that OOP payment is associated with household financial burdens [43]. Additionally, the study revealed an association between OOP payments influence on hospital attendance and difficulty in making OOP payments as survey respondents who indicated that OOP payments influenced their hospital attendance were five times more likely to have difficulty in making OOP payments compared to those for whom OOP payments had no influence on their hospital attendance.

Based on results from this study, survey respondents who had an alternative method of treating hypertension such as herbs were twice more likely to have a difficulty in making OOP payments than those who did not have any other method of treating hypertension apart from scheduled hospital visits. Research has reported that alternative forms of treatment such as home treatment has an effect on OOP payments [44].

OOP payments influenced the hospital attendance of 20.6% of the survey respondents as confirmed by research in Taiwan which indicated that prospective patients forego needed treatment and preventive care services due to high OOP expenditure resulting in negative clinical consequences [34]. This is also consistent with a similar study in Iran which reported that patients forego healthcare due to OOP payments [45].

A number of hypertensive patients recruited for the study admitted that they had alternative means of managing their condition with the recurring alternative being the use of local herbs. The study also revealed that 9% of the study respondents reported that OOP payments resulted in their conditions becoming worsened which is corroborated by a study which revealed that high OOP payments on prescription drugs is related with a higher rate of adverse outcomes and

hospital emergency unit visits among the aged and welfare beneficiaries [34]. Research suggests that while some OOP payments are related to essential medical treatments, certain other OOP payments are not dire for improving health status [34]. An examination of poverty consequences of healthcare spending indicates an upward trend of medical impoverishment resulting in great concern for policy makers and other stakeholders [39].

## Strengths and limitations

This was a comprehensive study that showed the prevalence of OOP payments, reasons for such payments and the self-reported effects of OOP payments among hypertensive patients who are subscribed to the NHIS. The current study provides useful insights based on the individuals' perspectives and adds to existing literature on OOP payments.

Our study, however, has some limitations which must be acknowledged. We employed a cross-sectional study design which did not demonstrate the cause-and-effect relationship between the dependent and independent variables but only determined associations. Additionally, this type of study only examined the current situation and may ignore changes over time. Also, our study did not distinguish between authorised and unauthorised payments. While we clarified the operational definition of payments during data collection, we acknowledge that self-reported payments may still be subject to recall bias and variability in interpretation. This could lead to misinterpretation of the data and as such, results should be interpreted with caution.

Further, the article does not fully assess the impact of OOP payments on the economic situation of households, especially poor households, which is essential to achieving the SDGs. Furthermore, the sample was drawn from only the Volta Region, which may not be representative of the entire country. Similarly, our findings were based on self-reported data which may lead to inaccuracies due to certain biases such as recall and social desirability. To address this, respondents were assured of anonymity and confidentiality. The facility-based nature of the study also minimized recall bias since respondents were interviewed immediately after having accessed health services. Sensitivity analyses were not conducted for the logistic regression models due to the distribution of the outcome variables.

These limitations should be considered when interpreting the findings and highlight opportunities for further research to address these gaps.

## Conclusion

OOP payments have financial consequences on households and with a remarkably high burden of OOP payments among hypertensive patients subscribed to the NHIS as observed by our study, there is the need to remain vigilant in ensuring that the targets of SDG 3 are attained in Ghana. To address the issue of payment for drugs covered under NHIS, the National Health Insurance Authority (NHIA) must ensure that drugs and services which are covered by the NHIS are available to all subscribers. Specifically essential medicines for controlling CNCDs must be made available as mandated under the NHIS and made accessible to those who need them in line with SDG 3.4. Based on this, the NHIA must address periodic drug stock-outs of medicines which hospitals face to ensure that healthcare seekers, especially hypertensive patients, have access to the right medications for their conditions. Periodic monitoring at health facilities must be undertaken by the NHIA to ensure that the scheme's mandate of attaining UHC in Ghana is achieved and any loopholes addressed. The government through the Ministry of Health must strengthen the NHIS, which is the tool for attaining UHC, to ensure that it delivers its mandate. It is also recommended that studies which determine the differences in OOP payments between insured and uninsured patients be conducted to estimate the level of financial protection the NHIS affords its active members.

## Supporting information

**S1 File. Dataset_outofpocketpayments.**
(XLSX)

## Author contributions

**Conceptualization:** Angela Nana Esi Ackon, Hubert Amu, Martin Amogre Ayanore.

**Formal analysis:** Angela Nana Esi Ackon, Hubert Amu, Martin Amogre Ayanore.

**Methodology:** Angela Nana Esi Ackon, Hubert Amu, Martin Amogre Ayanore.

**Supervision:** Hubert Amu, Martin Amogre Ayanore.

**Writing – original draft:** Angela Nana Esi Ackon, Hubert Amu, Martin Amogre Ayanore.

**Writing – review & editing:** Angela Nana Esi Ackon, Hubert Amu, Martin Amogre Ayanore.

## Acknowledgments

The completion of this work was possible thanks to the support of several people, to whom we take this opportunity to express our gratitude. Thus, we would like to express our gratitude to the management of the Volta Regional Hospital, Hohoe and the staff of the hypertensive clinic. We would also like to thank all the respondents for their time.

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
