## [Decision Letter · Decision Letter 0]

25 Apr 2025

PONE-D-25-08305Out-of-pocket payments and associated factors among hypertensive patients insured under the National Health Insurance Scheme in a referral hospital, GhanaPLOS ONE

Dear Dr. Ackon,

Thank you for submitting your manuscript to PLOS One. Firstly, we would like to apologize for the delay in processing your manuscript. It has been exceptionally difficult to secure reviewers to evaluate your study. We have now received one completed review, which is available below. The reviewer has raised significant scientific concerns about the study that need to be addressed in a revision.

Please note that we have only been able to secure a single reviewer to assess your manuscript. We are issuing a decision on your manuscript at this point to prevent further delays in the evaluation of your manuscript. Please be aware that the editor who handles your revised manuscript might find it necessary to invite additional reviewers to assess this work once the revised manuscript is submitted. However, we will aim to proceed on the basis of this single review if possible.

We look forward to receiving your revised manuscript.

Kind regards,

Miquel Vall-llosera Camps

Senior Staff Editor

PLOS ONE

2. We note that Figure 1 in your submission contain [map/satellite] images which may be copyrighted. All PLOS content is published under the Creative Commons Attribution License (CC BY 4.0), which means that the manuscript, images, and Supporting Information files will be freely available online, and any third party is permitted to access, download, copy, distribute, and use these materials in any way, even commercially, with proper attribution. For these reasons, we cannot publish previously copyrighted maps or satellite images created using proprietary data, such as Google software (Google Maps, Street View, and Earth). For more information, see our copyright guidelines: http://journals.plos.org/plosone/s/licenses-and-copyright.

Reviewers' comments:

Reviewer's Responses to Questions

**Comments to the Author**

1. Is the manuscript technically sound, and do the data support the conclusions?

Reviewer #1: Partly

2. Has the statistical analysis been performed appropriately and rigorously? 

Reviewer #1: Yes

3. Have the authors made all data underlying the findings in their manuscript fully available?

Reviewer #1: Yes

4. Is the manuscript presented in an intelligible fashion and written in standard English?

Reviewer #1: No

5. Review Comments to the Author

Reviewer #1: Dear author, thank you for your manuscript. The following comments are sent to enrich your work

It is better to focus more on the main challenge in the introduction and avoid additional explanations. State the main reason for choosing this topic and its importance.

State the time frame of the research.

Abbreviation for out-of-pocket payments is OOP not OPP.

The use of a cross-sectional method may not be able to show the cause-and-effect relationship between factors related to out-of-pocket payments. This type of study only examines the current situation and may ignore changes over time

This article does not provide a precise definition of "out-of-pocket payments" and does not distinguish between authorized and unauthorized payments. This could lead to misinterpretation of the data.

The sample was drawn from only three regions, which may not be representative of the entire country. This makes it difficult to generalize the results to other regions of Ghana.

The article does not fully assess the impact of out-of-pocket payments on the economic situation of households, especially poor households, which is essential to achieving the Sustainable Development Goals.

The article does not provide a precise definition of “out-of-pocket payments” and does not distinguish between authorized and unauthorized payments. This can lead to misinterpretation of the data.

Although the article provides recommendations, it does not provide practical solutions for implementing these recommendations.

6. PLOS authors have the option to publish the peer review history of their article (what does this mean?). If published, this will include your full peer review and any attached files.

Reviewer #1: No

---

## [Author Response · Author response to Decision Letter 1]

28 May 2025

University of Health and Allied Sciences

Fred N. Binka School of Public Health

PMB 31, Ho

May 28, 2025

The Editor-in-Chief

Plos One

Dear Dr. Camps,

Subject: Response to Reviewers for Manuscript [PONE-D-25-08305]

I sincerely appreciate the time and effort that you and the reviewer have invested in evaluating our manuscript, titled “Out-of-pocket payments and associated factors among hypertensive patients insured under the National Health Insurance Scheme in a referral hospital, Ghana”. We are grateful for the constructive feedback, which has helped us improve the quality and clarity of our work.

Below, we provide detailed responses to the editor and reviewer’s comments. We have carefully revised the manuscript to address all concerns and believe that the updated version strengthens our study.

Response: Thank you for pointing this out. We have thoroughly reviewed the journal’s submission guidelines and revised the paper accordingly. This includes adjustments to section headings, line spacing, abstract format and other formatting specifications.

2. We note that Figure 1 in your submission contain [map/satellite] images which may be copyrighted. We require you to either (1) present written permission from the copyright holder to publish these figures specifically under the CC BY 4.0 license, or (2) remove the figures from your submission.

Response: Thank you for pointing this out. We have removed the figure from the paper to avoid any copyright infringement.

Reviewer #1 comments:

1. Comment: It is better to focus more on the main challenge in the introduction and avoid additional explanations. State the main reason for choosing this topic and its importance.

Response: Thank you for this insightful comment. We appreciate your points on the need to be concise and specific. To improve clarity and focus, we have made major edits for conciseness while retaining key references which directly support the rationale for the research. We believe that this revision strengthens your reading experience by leading to the main reason for choosing this topic and its importance more quickly.

2. Comment: State the time frame of the research.

Response: Thank you for pointing this out. We have revised the manuscript to include the time frame of the research (See Pages 2, 7 & Lines 25, 116)

3. Comment: Abbreviation for out-of-pocket payments is OOP not OPP.

Response: Thank you for noting this. We have revised the entire manuscript to reflect OOP as the appropriate abbreviation for out-of-pocket and not OPP.

4. Comment: The use of a cross-sectional method may not be able to show the cause-and-effect relationship between factors related to out-of-pocket payments. This type of study only examines the current situation and may ignore changes over time.

Response: Thank you for this important point. We agree that cross-sectional studies are not able to determine causality and do not consider changes overtime. We have therefore acknowledged that our adoption of this study design is a key limitation of our study. This acknowledgement enhances the integrity of the paper.

5. Comment: This article does not provide a precise definition of "out-of-pocket payments" and does not distinguish between authorized and unauthorized payments. This could lead to misinterpretation of the data.

Response: Thank you for this insightful comment. We agree that the lack of a clear distinction between authorized and unauthorized out-of-pocket payments may lead to a misrepresentation of our findings. We have therefore acknowledged this as an important limitation of our study. This admission strengthens the transparency and credibility of the paper.

6. Comment: The sample was drawn from only three regions, which may not be representative of the entire country. This makes it difficult to generalize the results to other regions of Ghana.

Response: Thank you for this insightful comment. We agree that our study may not be generalizable to the other regions of Ghana. As such, we have acknowledged this as a major limitation of our study.

7. Comment: The article does not fully assess the impact of out-of-pocket payments on the economic situation of households, especially poor households, which is essential to achieving the Sustainable Development Goals.

Response: Thank you for this important point. We agree that our study did not comprehensively evaluate the impact of out-of-pocket payments on the economic situation of households. We have therefore acknowledged this as a key limitation of our study which we hope strengthens the transparency of the paper.

8. Comment: Although the article provides recommendations, it does not provide practical solutions for implementing these recommendations.

Response: Thank you for noting this. We have expanded our recommendations to include the key stakeholders responsible for implementing the recommendations we gave.

We hope that these revisions satisfactorily address the editor and reviewer’s concerns. We appreciate the opportunity to improve our manuscript and look forward to your feedback.

Sincerely,

Angela Nana Esi Ackon

angela.aan2009@gmail.com

---

## [Decision Letter · Decision Letter 1]

20 Feb 2026

PONE-D-25-08305R1Out-of-pocket payments and associated factors among hypertensive patients insured under the National Health Insurance Scheme in a referral hospital, GhanaPLOS One

Dear Dr. Ackon,

Thank you for submitting your manuscript to PLOS ONE. After careful consideration, we feel that it has merit but does not fully meet PLOS ONE’s publication criteria as it currently stands. Therefore, we invite you to submit a revised version of the manuscript that addresses the points raised during the review process.

We look forward to receiving your revised manuscript.

Kind regards,

Charles C Ezenduka, PhD

Academic Editor

PLOS One

Journal Requirements:

Additional Editor Comments:

The last reviewer has raised a significant and valid concerns that require authors' attention and responses to enhance quality of the manuscript to meet publication criteria.

Hence, authors are requested to respond to these comments based on a minor essential revision

Reviewers' comments:

Reviewer's Responses to Questions

**Comments to the Author**

1. If the authors have adequately addressed your comments raised in a previous round of review and you feel that this manuscript is now acceptable for publication, you may indicate that here to bypass the “Comments to the Author” section, enter your conflict of interest statement in the “Confidential to Editor” section, and submit your "Accept" recommendation.

Reviewer #1: All comments have been addressed

Reviewer #2: All comments have been addressed

Reviewer #3: All comments have been addressed

2. Is the manuscript technically sound, and do the data support the conclusions?

Reviewer #1: Yes

Reviewer #2: Yes

Reviewer #3: Yes

3. Has the statistical analysis been performed appropriately and rigorously? 

Reviewer #1: Yes

Reviewer #2: Yes

Reviewer #3: Yes

4. Have the authors made all data underlying the findings in their manuscript fully available?

Reviewer #1: Yes

Reviewer #2: Yes

Reviewer #3: Yes

5. Is the manuscript presented in an intelligible fashion and written in standard English?

Reviewer #1: Yes

Reviewer #2: Yes

Reviewer #3: Yes

6. Review Comments to the Author

Reviewer #1: Dear Editor,

Hello

I confirm that all the reviewer comments provided to the authors have been fully addressed and incorporated into the revised version of the manuscript. The authors have responded appropriately to each point, and the revisions made adequately reflect the feedback previously given.

Reviewer #2: The revised manuscript provides a timely and policy-relevant analysis of out-of-pocket payments among insured hypertensive patients in Ghana, addressing an important gap in the universal health coverage discourse. The methodology is appropriate, with clear use of cross-sectional design, adequate sample size, and robust multivariable logistic regression analysis. The findings are clearly presented and demonstrate significant associations with meaningful public health implications. The discussion appropriately contextualizes the results within SDG targets and national insurance policy objectives. I recommend acceptance of the manuscript in its current form.

Reviewer #3: General comment

The authors have clearly engaged with the prior reviewer comments and have added explicit limitations, clarified timeframe, corrected terminology, and expanded recommendations. The manuscript is scientifically coherent and addresses an important health-financing question. However, several editorially critical and methodologically relevant issues remain unresolved, including internal inconsistencies, reporting clarity, sampling description accuracy, and analytical interpretation.

Specific comments

1. While the authors state that all instances of OPP were replaced with OOP, the revised manuscript still contains multiple residual “OPPsOOP” artifacts, indicating incomplete editing and typesetting inconsistencies (visible in conclusions and limitations sections).

2. The limitation about not distinguishing authorised vs unauthorised payments is acknowledged, which is good, but the manuscript still does not operationally define how respondents interpreted OOP payments during data collection. This raises interpretability concerns about whether co-payments and informal payments, alongside drug purchases outside facility were included. In essence, the limitation statement remains insufficient as readers need clearer operational framing in the Methods.

3. While the limitation about single-region sampling is properly acknowledged, the reviewer response incorrectly states the sample was from three regions whereas the manuscript clearly describes one facility in the Volta Region. This mismatch should be corrected for consistency.

4. While stakeholder identification has been added to the recommendations, most of the policy recommendations remain generic, with limited linkage to study findings. I recommend that the authors tie each recommendation directly to a specific empirical result.

5. The manuscript states: sample size target = 422 (after non-response adjustment), and actual sample recruited = 389. However, there is no explanation for this discrepancy. This reporting gap could affect reproducibility and transparency. In addition, the randomisation description suggests systematic daily recruitment rather than true simple random sampling, and while a potential repeat-visit bias mitigation is described, it was not validated.

6. While the authors acknowledge that outcome variables rely entirely on self-report, this could be better contextualised along recall bias, misclassification and social desirability bias risks.

7. While the Logistic regression seems appropriate, some missing essentials include variable selection criteria, model diagnostics, multicollinearity checks, and sensitivity analysis. These elements of model transparency would improve rigour.

8. In result interpretation, the prevalence estimate (97.2%) is extremely high relative to cited Ghanaian literature. While the discussion attributes this to sample homogeneity, this explanation is not empirically demonstrated. Thus, the conclusion risks overstating generalisable burden.

9. Several issues require correction as multiple sections contain duplicated or corrupted terms (e.g., OPPsOOP). For instance, percent denominators are sometimes unclear, there are overlapping categories, and Fisher vs Chi-square labelling seem inconsistent.

10. Also, there are some logical phrasing issues such as “findings are important and generalisable”. This is contradicted by the already stated sampling limitations. Again, the claim that findings are generalisable is not sufficiently justified given the single facility, region-specific context, and older patient skew. A revision should reflect analytic relevance, not population generalisability.

11. The revised manuscript does not report any formal robustness or sensitivity analyses beyond the primary logistic regression model. Including some basic model diagnostics or alternative specifications would strengthen confidence in the findings. Given the very high prevalence of OOP payments and reliance on self-reported measures, demonstrating that the main associations are not sensitive to modelling choices would enhance the methodological transparency and credibility of the results.

7. PLOS authors have the option to publish the peer review history of their article (what does this mean?). If published, this will include your full peer review and any attached files.

Reviewer #1: No

Reviewer #2: **Yes:** Dr. Ragni Kumari

Reviewer #3: No

---

## [Author Response · Author response to Decision Letter 2]

15 Apr 2026

I sincerely appreciate the time and effort that you and the reviewers have invested in evaluating our manuscript, titled “Out-of-pocket payments and associated factors among hypertensive patients insured under the National Health Insurance Scheme in a referral hospital, Ghana”. We are grateful for the constructive feedback, which has helped us improve the quality and clarity of our work.

Below, we provide detailed responses to the editor and reviewers’ comments. We have carefully revised the manuscript to address all concerns and believe that the updated version strengthens our study.

Reviewer #1: Dear Editor,

Hello

I confirm that all the reviewer comments provided to the authors have been fully addressed and incorporated into the revised version of the manuscript. The authors have responded appropriately to each point, and the revisions made adequately reflect the feedback previously given.

Reviewer #2: The revised manuscript provides a timely and policy-relevant analysis of out-of-pocket payments among insured hypertensive patients in Ghana, addressing an important gap in the universal health coverage discourse. The methodology is appropriate, with clear use of cross-sectional design, adequate sample size, and robust multivariable logistic regression analysis. The findings are clearly presented and demonstrate significant associations with meaningful public health implications. The discussion appropriately contextualizes the results within SDG targets and national insurance policy objectives. I recommend acceptance of the manuscript in its current form.

Reviewer #3: General comment

The authors have clearly engaged with the prior reviewer comments and have added explicit limitations, clarified timeframe, corrected terminology, and expanded recommendations. The manuscript is scientifically coherent and addresses an important health-financing question. However, several editorially critical and methodologically relevant issues remain unresolved, including internal inconsistencies, reporting clarity, sampling description accuracy, and analytical interpretation.

Specific comments

1. Comment: While the authors state that all instances of OPP were replaced with OOP, the revised manuscript still contains multiple residual “OPPsOOP” artifacts, indicating incomplete editing and typesetting inconsistencies (visible in conclusions and limitations sections).

Response: Thank you for pointing this out. We have thoroughly revised the entire manuscript and expunged all residual artifacts. We have also ensured consistency in editing across all sections.

2. Comment: The limitation about not distinguishing authorised vs unauthorised payments is acknowledged, which is good, but the manuscript still does not operationally define how respondents interpreted OOP payments during data collection. This raises interpretability concerns about whether co-payments and informal payments, alongside drug purchases outside facility were included. In essence, the limitation statement remains insufficient as readers need clearer operational framing in the Methods.

Response: Thank you for this insightful comment. We have revised the methods to clearly detail how respondents were instructed to interpret payments. Our study however excludes payments made outside of the facility since we sought to examine the out-of-pocket payments made which would have ordinarily been covered by the NHIS. In our revision, we have strengthened the operational definition of payments in the methods section. We believe that these revisions would enhance transparency (See Page 9, Lines 164-173).

3. Comment: While the limitation about single-region sampling is properly acknowledged, the reviewer response incorrectly states the sample was from three regions whereas the manuscript clearly describes one facility in the Volta Region. This mismatch should be corrected for consistency.

Response: Thank you for this important observation. We acknowledge that our sample was drawn from one facility in the Volta region and hence our response to the previous review comment should have clarified the absence of any mention of three regions in our manuscript.

4. Comment: While stakeholder identification has been added to the recommendations, most of the policy recommendations remain generic, with limited linkage to study findings. I recommend that the authors tie each recommendation directly to a specific empirical result.

Response: Thank you for this insightful comment. We have revised the recommendations to relate it more closely with the results to enhance their practical relevance.

5. Comment: The manuscript states: sample size target = 422 (after non-response adjustment), and actual sample recruited = 389. However, there is no explanation for this discrepancy. This reporting gap could affect reproducibility and transparency. In addition, the randomisation description suggests systematic daily recruitment rather than true simple random sampling, and while a potential repeat-visit bias mitigation is described, it was not validated.

Response: Thank you for this insightful comment. We have revised the methods to clarify that the discrepancy reflects practical recruitment constraints rather than methodological issues. We however successfully recruited 393 respondents but 4 were not included in the analysis due to missing data. Also, we have now revised the methods section to clarify the process of selecting participants. Regarding the mitigation of potential repeat-visit bias, we agree that our initial description lacked validation and we have revised our limitations to clearly acknowledged this limitation. We believe these revisions strengthen methodological clarity of the manuscript.

6. Comment: While the authors acknowledge that outcome variables rely entirely on self-report, this could be better contextualised along recall bias, misclassification and social desirability bias risks.

Response: Thank you for this observation. We have revised the limitations to address the possibilities of recall bias, misclassifications and social desirability bias. We have also clarified in our methods and limitations sections and provided a fuller account of both the risks and the mitigation strategies employed. Firstly, that questionnaires were administered as exit interviews, immediately after respondents received treatment which reduced the risk of recall bias. Again, we have noted that we tried to reduce misclassification by providing detailed breakdown of possible health services that may have resulted in payments in the questionnaire and with regards to social desirability bias, we minimised this by conducting the interviews privately.

7. Comment: While the Logistic regression seems appropriate, some missing essentials include variable selection criteria, model diagnostics, multicollinearity checks, and sensitivity analysis. These elements of model transparency would improve rigour. Response: Thank you for this important comment. We could not conduct sensitivity analysis due to the distribution of the outcome variables which limited the potential for meaningful variation across alternative model specifications. We have expanded our limitations to acknowledge this (See Page 27, Lines 494-495).

8. Comment: In result interpretation, the prevalence estimate (97.2%) is extremely high relative to cited Ghanaian literature. While the discussion attributes this to sample homogeneity, this explanation is not empirically demonstrated. Thus, the conclusion risks overstating generalisable burden.

Response: Thank you for this important comment. We have made revisions in our manuscript to ensure that we remain within the limits of our evidence. We have thus accurately presented results, expanded the limitations to fully explain methodological and contextual constraints and revised our conclusions to avoid overstating implications to enhance the credibility of the manuscript.

9. Comment: Several issues require correction as multiple sections contain duplicated or corrupted terms (e.g., OPPsOOP). For instance, percent denominators are sometimes unclear, there are overlapping categories, and Fisher vs Chi-square labelling seem inconsistent.

Response: Thank you for this insightful comment. We have revised the manuscript and corrected all instances of duplication or corrupted text. We have clarified the denominators used for each percentage, ensuring consistency and transparency across tables and narrative results. These corrections improve clarity, accuracy, and consistency throughout the manuscript.

10. Comment: Also, there are some logical phrasing issues such as “findings are important and generalisable”. This is contradicted by the already stated sampling limitations. Again, the claim that findings are generalisable is not sufficiently justified given the single facility, region-specific context, and older patient skew. A revision should reflect analytic relevance, not population generalisability.

Response: Thank you for this important observation. We have made major revisions in our manuscript to ensure that we avoid overstating implications of our findings. We believe this would enhance the credibility of the manuscript.

11. Comment: The revised manuscript does not report any formal robustness or sensitivity analyses beyond the primary logistic regression model. Including some basic model diagnostics or alternative specifications would strengthen confidence in the findings. Given the very high prevalence of OOP payments and reliance on self-reported measures, demonstrating that the main associations are not sensitive to modelling choices would enhance the methodological transparency and credibility of the results.

Response: Thank you for this important observation. Because our study reported a very high prevalence of OOP payments, the scope for meaningful sensitivity analysis was limited given that for such a skewed distribution, sensitivity analyses across subsamples or alternative specifications would not materially change the results, as the imbalance in outcome frequencies constrains model variation. We have acknowledged this in our limitations and noted that future research could consider alternative approaches if outcomes with greater variability are available.

We hope that these revisions satisfactorily address the editor and reviewers’ concerns. We appreciate the opportunity to improve our manuscript and look forward to your feedback.

Sincerely,

Angela Nana Esi Ackon

angela.aan2009@gmail.com

---

## [Editor Report · Decision Letter 2]

24 Apr 2026

Out-of-pocket payments and associated factors among hypertensive patients insured under the National Health Insurance Scheme in a referral hospital, Ghana

PONE-D-25-08305R2

Dear Dr. Ackon,

We’re pleased to inform you that your manuscript has been judged scientifically suitable for publication and will be formally accepted for publication once it meets all outstanding technical requirements.

Kind regards,

Charles C Ezenduka, PhD

Academic Editor

PLOS One
---

## [Editor Report · Acceptance letter]

PONE-D-25-08305R2

PLOS One

Dear Dr. Ackon,

I'm pleased to inform you that your manuscript has been deemed suitable for publication in PLOS One. Congratulations! Your manuscript is now being handed over to our production team.

Kind regards,

on behalf of

Dr. Charles C Ezenduka

Academic Editor

PLOS One